# Social capital, depressive symptomatology, and frailty among older adults in the western areas of China

Liqun Wang[1,2], Shufeng Xie[3], Xue Hu[4], Jiangping Li[1,2], Shulan He[1,2], Junling Gao[5]*, Zhizhong Wang[4]*

1 Department of Epidemiology and Statistics, School of Public Health at Ningxia Medical University, Yinchuan, China, 2 Key Laboratory of Environmental Factors and Chronic Disease Control, Ningxia Medical University, Yinchuan, China, 3 General Hospital of Ningxia Medical University, Yinchuan, China, 4 Department of Epidemiology and Statistics at School of Public Health of Guangdong Medical University, Dongguan, China, 5 Department of Health Education at School of Public Health, Fudan University, Shanghai, China

* jlgao@fudan.edu.cn (JG); wzhzh_lion@126.com (ZW)

**Data Availability Statement:** All relevant data are within the paper and its Supporting Information files.

## Abstract

We aimed to explore the relationship between social capital (SC) and frailty, and the mediation role of depressive symptoms in this relationship. A cross-sectional study among 2,591 older adults aged ≥60 years old was conducted from September 2020 to May 2021. SC, depressive symptoms, and frailty were measured using the social capital scale, the 9-item patient health questionnaire, and the FRAIL scale, respectively. The mediation model was tested by Bootstrap PROCESS. After controlling for socio-demographical covariates, the SC was negatively correlated with frailty ($r = -0.07$, $P = 0.001$), and depressive symptomatology ($r = -0.08$, $P<0.001$); while the depressive symptomatology was positively correlated with frailty ($r = 0.33$, $P<0.001$). Logistic regression results showed that SC was associated with a lower risk of frailty ($OR = 0.94$; 95% $CI$: 0.92–0.97; $P<0.001$). Depressive symptomatology partially mediated (explained 36.4% of the total variance) the association between SC and frailty. Those findings suggest that SC may protect older adults from frailty by reducing depressive symptoms. Prevention and intervention implications were also discussed.

## Introduction

Over the past decades, population aging has been observed worldwide, consequently, an increased prevalence of age-related disease caused a huge disease burden and challenged the healthcare system [1]. Frailty is theoretically defined as a clinically recognizable state of increased vulnerability resulting from the aging-associated decline in reserve and function across multiple physiologic systems such that the ability to cope with everyday or acute stressors is comprised [2]. Studies have found this multidimensional clinical geriatric syndrome was associated with several adverse outcomes such as falls [3, 4], depression [5], disability [6], dementia [7], and excessive mortality [8]. Frailty is reversible at an early stage indicating efforts on reducing risk factors will help develop targeted interventions to prevent the frailty in advance.

The determinants of frailty include genetic [9, 10] and environmental factors (education, lifestyle, nutrition, etc) [11–13]. Studies have reported that depressive symptomology is an

**Funding:** this work was supported by the National Natural Science Foundation of China (grant number 81860599)". The funder had no role in study design, data collection and analysis, decision to publish, or preparation of the manuscript.

**Competing interests:** The authors have declared that no competing interests exist.

independent psychological predictor of frailty [14, 15]. A meta-analysis pooled several longitudinal studies found that the risk of new occurred frailty in women (aged 65 to 79 years) with the highest level of depressive symptomatology ($\geq 5$ on the six-item CES-D) was significantly increased compared to women with mild or no depressive symptomatology [5, 16].

Social capital (SC) has been widely recognized as a positive social determinant of physical and mental health [17, 18]. It is a characteristic of social life, including interpersonal trust, norms of reciprocity, mutual aid, and social involvement [19–21]. Studies have found that both community-level SC and individual-level SC are negatively associated with the risk of frailty [22, 23]. Living in a community with rich civic participation, such as engagement in social activities, was associated with a lower risk of frailty onset among older adults (OR = 0.94) [23]. However, to our knowledge, little research intends to explain how a higher level of SC reduces the risk of frailty. One study indicates that health-promoting lifestyles partially mediated the association between social capital and frailty [24], suggesting that health-promoting lifestyles targeted interventions may reduce the risk of frailty among older adults. Afterward, SC has been correlated with depression [25]. Quite a few studies suggested that older people with higher social capital had a lower risk of occurring depression [26–28].

China is facing a rapid population aging currently, the proportion of the population aged 60 years or older increased to 18.7% (+5.44% compared to 2010) in the year 2020 [29]. A previous study reported that the overall prevalence of frailty was 9.9% in Chinese older adults [30]. To develop effective prevention of frailty in older adults from the SC perspective, the potential mediating role in the relationship between SC and frailty ought to be understood. Prior work has found that SC can indirectly affect health by reducing the risk of depression [31]. However, it remains unknown whether the mediating role of depression can be generalized to the relationship between SC and frailty. Accordingly, the main objective of the current study is to explore the relationship between SC and frailty among older adults and to test the mediation effect of depressive symptomatology on this relationship.

## Methods

### Participants

Participants in the present study were acquired from Ningxia province in mainland China between September 2020 and May 2021 during the Health Examination for Urban and Rural Residents program. Eight communities were selected from 3 districts (Litong, Xingqing, and Dawukou district). Permanent residents staying more than 6 months aged over 60 years were eligible to participate; excluding those who were unable to complete the survey due to vision and hearing disabilities, or a history of serious physical illness. Totally, 2,681 community residents enrolled and finished the face to face survey, of them, 64 were excluded due to the missing value of frailty scale (n = 42), and PHQ-9 (n = 22), and 26 aged less than 60 were excluded. Thus, 2,591 subjects (S1 Table) were included in the final analysis.

The research procedures of this study involving human participants conformed with the ethical standards of the Institutional Review Board of the Ningxia Medical University (Nos. 2018115) and with the 1964 Helsinki Declaration and the later amendments or similar ethical standards. All participants signed a written informed consent form before the survey including some of illiterate participants (the investigators read to them and they can sign their names).

### The field process

Face-to-face interviews were conducted by trained lay health professionals in the primary healthcare settings, all the potential participants were invited to join the study when they attend the pay-free annual health examination offered by the local governmental public health

service. The participants agreed to be read each question by our investigator and then their answers were recorded. The survey lasted around 40 minutes. The finished questionnaire was double-checked immediately by two separate supervisors in the field.

## Measures

**Frailty.**   The FRAIL scale was employed to assess the frailty status [32]. This five-item self-rating scale combines components of a functional model, a cumulative model, and a bio-frailty model [33]. Items including weight loss (defined as unintentional weight loss over the past year equal to or greater than 5% or more than 4.5 kilograms), fatigue (in the past month, have you often felt tired), slowness (Can you walk continuously for 10 minutes or 400 meters), illness (suffering from 5 or more chronic diseases), and endurance (Climb up ten steps or one floor without a break or aid of any kind), with a total score of 5 points. Individual scoring $\geq 3$ points was defined as frailty. The scale has good construct validity and reliability.

**Social capital.**   SC that covered social cohesion, and social interaction is assessed using two separate instruments. Social cohesion is assessed using the Neighborhood Scales developed by Mujahid [34], which consists of 4 items, and the social interaction scale consists of five items. The sum of the 9-item (using 5-point Likert scoring ranging from 1 = "strongly disagree" to 5 = "strongly agree") is the total score. The higher the score indicates the greater the social capital of the participant. Both instruments are widely used to assess the social capital of Chinese adults [35, 36], the Cronbach's alpha in this sample is 0.86.

**Depressive symptomology.**   The Patient Health Questionnaire 9 (PHQ-9) was used to measure depressive symptomology during the past two weeks [37]. PHQ-9 is a self-rating questionnaire that consists of nine depression criteria from the DSM-IV [38]. Each item ranged from 0 to 3 (0 = none at all, 1 = a few days, 2 = more than half the time, 3 = almost daily) that resulting in a total score ranging from 0 to 27 [39]. Generally, a scoring threshold $\geq 10$ is defined as significant depressive symptoms [40]. The Cronbach's alpha in this sample is 0.88.

**Covariates.**   A host of studies have made of quite a few demographic characteristics and lifestyle and health-related behaviors that play a parament role in the development of frailty. In this study, the selected covariates include age (continuous variable), gender (0 = female, 1 = male), marital status (0 = married, 1 = Widowed or divorced), Living area (0 = rural, 1 = urban), educational attainment (1 = illiterate, 2 = primary, 3 = junior, 4 = senior and above), occupation (0 = non-farmer, 1 = farmer), family income ($0 \geq 2000$ RMB, $1 < 2000$ RMB), smoking (defined as having smoked 100 cigarettes in one's lifetime and currently smoking cigarettes) [41], alcohol use (defined as have often have a drink, 0 = never, 1 = once a month or less, 2 = 2–4 times a month, 3 = 2–3 times a week, 4 = 4 times a week or more), physical activity (defined as how many times per week you engage in moderate-intensity (increased heart rate, breathing rate, slightly sweating, 0 = never, 1 = 1–2 times, 2 = more than 3 times)), general health status (defined as how do you think about your general health, categorized individuals as good, moderate, bad), living alone (0 = no 1 = yes), poor sleep quality (according to the Pittsburgh Sleep Quality Scale, with a score of 7 or more indicating poor sleep quality) [42].

**Statistical analysis.**   Continuous and categorical variables were expressed as mean (SD) and numbers (percentages) respectively. Demographic characteristics between the two samples were analyzed by Chi-squared test or Student's t test. A correlation matrix was created using partial correlations under controlling for age, gender, living area, educational attainment, marital status, occupation, family income, smoking, alcohol use, physical activity, general health status, living alone, and poor sleep quality. Logistic regression model was employed to examine the association between SC and frailty, and odds ratios (ORs) and 95% confidence intervals (95% CIs) were calculated under controlling for covariates. We looked at the role of

depressive symptoms in mediating the relationship between SC and frailty using the bootstrap methods of PROCESS developed by Hayes [43]. Having a bias-corrected bootstrap confidence interval that does not contain 0 indicates that there is a mediation effect [44]. Statistical significance was set at $P < 0.05$. All the analyses were performed using the Statistical Package for Social Sciences version 24.0 (SPSS Inc., Chicago, Illinois).

## Results

### The characteristics of the participants

As shown in **Table 1**, the participants with an average age of 72.1 (SD = 5.7), and an average social capital score of 29.2 (SD = 5.9). The average score on the depressive symptom was 5.1 (SD = 5.0), with 47.7% of subjects classified as highly social capital, and 24.0% met the criteria of frailty.

### The binary correlation matrix

The partial correlation matrix showed in **Table 2**. After controlling for covariate variables, the SC was negatively correlated with frailty (r = -0.07, P = 0.001), the SC was also negatively

**Table 1. Demographic characteristics of the participants.**

| Variables | Total (N = 2,591) |
|---|---|
| Gender, male, n (%) | 1,184(45.7) |
| Residence, rural, n (%) | 1,265(48.8) |
| Marital status, n (%) | |
| Married | 2,083(80.4) |
| Widowed or divorced | 508(19.6) |
| Education level, n (%) | |
| Illiterate | 1,259(48.8) |
| Primary | 587(22.5) |
| Junior | 459(17.7) |
| Senior and above | 286(11.0) |
| Occupation, farmer, n (%) | 1,522(58.7) |
| Family income, <2000, n (%) | 1,465(56.5) |
| Smoking, n (%) | 354(13.7) |
| Alcohol use, n (%) | 242(9.3) |
| Physical activity, n (%) | |
| 0 | 946(36.5) |
| 1–2 | 269(10.4) |
| ≥3 | 1,376(53.1) |
| Self-assessed health status | |
| Good | 1,219(47.0) |
| Fair | 965(37.2) |
| Poor | 407(15.7) |
| Living alone, n (%) | 333(12.9) |
| Age, mean (SD), years | 72.1(5.7) |
| Depression, mean (SD) | 5.1(5.0) |
| Social capital, mean (SD) | 29.2(5.9) |
| Poor sleep quality, n (%) | 1,195(46.1) |
| Highly social capital, n (%) | 1,235(47.7) |
| Frailty, n (%) | 621(24.0) |

**Table 2. Correlation matrix[#] (n = 2,591).**

|  | Mean | SD | Depressive symptoms | Social Capital | Frailty |
|---|---|---|---|---|---|
| Depressive symptoms | 5.1 | 5.0 | 1 |  |  |
| Social Capital | 29.2 | 5.9 | -0.08** | 1 |  |
| Frailty | 1.9 | 0.9 | 0.33** | -0.07* | 1 |

**p<0.001

*p<0.05, SD = standard deviation

[#]After controlling for age, gender, residence, education level, marital status, occupation, family income, smoking, alcohol use, physical activity, general health status, living alone, poor sleep quality

correlated with depressive symptom (r = -0.08, P<0.001); the depressive symptom was positively correlated with frailty (r = 0.33, P<0.001).

## Logistic regression model

The logistic regression model is presented in **Table 3**. When controlling for covariates, the SC was associated with a lower risk of frailty (OR = 0.94; 95% CI: 0.92–0.97; P<0.001). The association between depressive symptom and frailty disappeared when adding the interaction between SC and depressive symptom (model 3), indicating that depressive symptom is a possible mediator.

## Mediation effect of depression on the relationship of SC and frailty

As shown in **Table 4**, after controlling for covariates, there is a significant mediation effect of depression on the relationship between SC and frailty. The total effect of SC on frailty was significant (total effect, β = -0.011; 95% CI = -0.017,– 0.005; P = 0.007). The estimated β (95% CIs and P value) of a significant indirect effect mediated by depression was -0.004 (95% CI = -1.006,–0.002 and P = 0.039). The mediation effect explained 36.4% (-0.004/-0.011) of the total variance.

## Discussion

The current study examined the association between SC, depressive symptoms, and frailty. As hypothesized, we found: (1) frailty was prevalent among older adults in northwest China; (2) SC was negatively related to depressive symptoms and frailty; depression was positively related to frailty; (3) depressive symptoms accounted for 36.4% of the total effect in the relationship

**Table 3. Logistic regression model for interaction between SC and depression on frailty (n = 2,591).**

| Variables | Model 1 | | Model 2 | | Model 3 | |
|---|---|---|---|---|---|---|
|  | P value | OR (95%CI) | P value | OR (95%CI) | P value | OR (95%CI) |
| SC | 0.005 | 0.98(0.96,0.99) | 0.043 | 0.98(0.96,0.99) | <0.001 | 0.94(0.92,0.97) |
| Depressive symptoms | <0.001 | 1.17(1.15,1.19) | <0.001 | 1.13(1.11,1.16) | 0.642 | 0.98(0.89,1.07) |
| SC × depressive symptoms | NA | NA | NA | NA | 0.001 | 1.005(1.002,1.008) |

Model l = SC or depressive symptoms separately; Model 2 = Model l + covariate variables (age, gender, residence, education level, marital status, occupation, family income, smoking, alcohol use, physical activity, general health status, living alone, poor sleep quality); Model 3 = Model 2 + interaction between SC and depression; SC: social capital; OR: Odds ratio; 95%CI: 95% confidence interval; NA: not apply

**Table 4. The mediating effect of depression on the relationship between SC and frailty \*.**

| Effect | | | | Bias-Corrected 95%CI | |
|---|---|---|---|---|---|
| | β | SE | P-value | Lower | Upper |
| Total effect | -0.011 | 0.003 | 0.007 | -0.017 | -0.005 |
| Indirect Effects | -0.004 | <0.001 | 0.039 | -0.006 | -0.002 |
| Direct Effects | -0.007 | 0.003 | 0.021 | -0.013 | -0.001 |

\*After controlling for age, gender, residence, education level, marital status, occupation, family income, smoking, alcohol use, physical activity, general health status, living alone, and poor sleep quality.

between SC and frailty. Those findings provide the primary evidence for the relationship between SC, depressive symptoms, and frailty among older Chinese adults, suggesting the possible mechanisms to elucidate and interpret how SC is related to lower risk of frailty.

In the current study, 24.0% of the older adults with frailty, the prevalence is lower than that found in Brazil (47.2%) [45] and Spain (27.3%) [46], while higher than that found in the east area of China (3.8%) [47]. The reasons for this phenomenon may be owing to different samples, geographic variation [48], the assessment instrument [49], and so on.

The elderly with high SC were 0.98 times less likely to be frail in this study. This result was in line with previous studies [24, 50–52]. The latest study demonstrated that social capital was associated with a reduced likelihood of frailty [24]. The possible mechanism may be that communicating with others can obtain emotional and social support and then buffer against physiological stress, thereby reducing the likelihood of frailty [24].

In line with prior studies [5, 53], we found depression was positively associated with frailty, older adults with depressive symptoms were 1.13 times more likely to be frailty. There are several possible explanations, first, it is reported that depression was associated with higher levels of inflammation and oxidative stress [54] which is believed one of the biological mechanisms of frailty in older adults [55]. On the other hand, depression can lead to unbalanced food intake and physical inactivity [56], which in turn results in frailty [5].

We also found that depressive symptoms mediated the association between SC and frailty, which offers additional evidence to appreciate the underlying mechanisms of the association between SC and frailty and may provide a special insight to prevent and intervene frailty among older adults via psychological path. Consistent with previous research [27], individuals with high level of SC are more likely to perceive a higher level of life satisfaction that may be against depressive symptoms [57]. The mechanism also may be that SC enhances the individuals' capacity to handle adverse life events and then further reduces depressive symptoms [58]. Hence, SC might relieve the frailty via decrease the depressive symptoms.

## Strength and limitations

Given China's rapidly aging population, frailty in older adults is increasingly becoming paramount. The present findings have relevance for understanding the mechanisms of how SC is linked with frailty. And provide primary evidence for implementing interventional strategies for frailty in minority areas. Several limitations were mentioned in our study; First, the cross-sectional design prevents examining the causal relationships between SC, depressive symptoms, and frailty, here needs longitudinal design to confirm the association in the future. Second, all surveys were self-reported, and that may have introduced information bias. Third, the sample is from one province in China, so the findings may not be generalizable to older adults in other countries and regions.

## Conclusion

Higher SC was associated with a reduced likelihood of frailty, and this association was partially mediated by depressive symptoms. The findings provide primary evidence for a better understanding of how SC is associated with frailty. Hence, increasing social capital (communicating more with others, fostering neighborhood relationships, etc.) to identify interventions for frailty directly or suppress depression via a psychological path to relieving frailty is essential.

## Supporting information

**S1 Table. Original data.**
(XLS)

## Acknowledgments

The authors sincerely thank all study participants and research staff that have contributed to this work.

## Author Contributions

**Conceptualization:** Liqun Wang, Zhizhong Wang.

**Data curation:** Liqun Wang, Shufeng Xie, Xue Hu, Jiangping Li, Shulan He, Junling Gao, Zhizhong Wang.

**Formal analysis:** Liqun Wang, Shufeng Xie, Junling Gao, Zhizhong Wang.

**Writing – original draft:** Liqun Wang, Zhizhong Wang.

**Writing – review & editing:** Junling Gao, Zhizhong Wang.

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
