## [Decision Letter · Decision Letter 0]

24 Jul 2023

PONE-D-23-09139Social capital, depressive symptomatology, and frailty among the older adults in the Western areas of ChinaPLOS ONE

Dear Dr. Wang,

Thank you for submitting your manuscript to PLOS ONE. After careful consideration, we feel that it has merit but does not fully meet PLOS ONE’s publication criteria as it currently stands. Therefore, we invite you to submit a revised version of the manuscript that addresses the points raised during the review process.

We look forward to receiving your revised manuscript.

Kind regards,

Wei-Min Chu

Academic Editor

PLOS ONE

Journal Requirements:

https://www.longdom.org/open-access-pdfs/a-confusing-general-term-frailty-should-be-organized-in-relation-with-frieds-criteria-for-frailty-locomotive-syndrome-mu.pdf

https://www.researchgate.net/publication/310021666_Social_capital_interventions_targeting_older_people_and_their_impact_on_health_A_systematic_review

https://bmcgeriatr.biomedcentral.com/articles/10.1186/s12877-022-02815-z

In your revision ensure you cite all your sources (including your own works), and quote or rephrase any duplicated text outside the methods section. Further consideration is dependent on these concerns being addressed.

4.Thank you for stating the following financial disclosure: 

"The authors received no specific funding for this work."

5. We are unable to open your Supporting Information file [database.sav]. Please kindly revise as necessary and re-upload.

Additional Editor Comments:

Please refer to reviewers' comments and revise the manuscript per request.

Reviewers' comments:

Reviewer's Responses to Questions

**Comments to the Author**

1. Is the manuscript technically sound, and do the data support the conclusions?

Reviewer #1: Partly

Reviewer #2: Partly

2. Has the statistical analysis been performed appropriately and rigorously? 

Reviewer #1: No

Reviewer #2: I Don't Know

3. Have the authors made all data underlying the findings in their manuscript fully available?

Reviewer #1: Yes

Reviewer #2: No

4. Is the manuscript presented in an intelligible fashion and written in standard English?

Reviewer #1: No

Reviewer #2: No

5. Review Comments to the Author

Reviewer #1: 1. Pay attention to the tense and grammar of the article in writing.

2. In Table 2, "After controlling for covariates, SC was negatively correlated with frailty (r=0.07, P=0. 001) and SC was also negatively correlated with depressive symptoms (r=-0.08, P<0.001)", the correlation coefficient seems to be too small for the authors to re-verify the statistical significance.

3. Line 190, r = -0.07 instead of 0.07.

4. The discussion section authors should focus on how depression plays a moderating role in social capital and debilitation.

Reviewer #2: 1. This is a cross-sectional study among 2,591 older adults aimed to explore the relationship between social capital (SC) and frailty. The logistic regression analysis showed that SC was associated with a lower risk of frailty; depressive symptomatology seemed partially mediated (explained 36.4% of the total variance) the association between SC and frailty.

2. Abstract: Line 44~45: ” SC was identified as a protective predictor of frailty in Chinese old(er) adults in this study…” The authors need to be very careful of making this conclusion because it just shows association but not causality. Does this conclusion overreach?

3. Participants: Line 100: Sample numbers were inconsistent. (2681-90 (42+22+26) = 2591)

4. Results: Line 199~203: Mediation effect of depression on the relationship of SC and frailty was not explained clearly enough. Is the statistical analysis adequate? Is there any possible confounder?

5. “Older adults” is better wording than “old adults”, it should be constantly the same all through the paper.

6. English editing and grammatical adjustment are required.

7. Please rearrange Table 1 and Table 3 to be shorter and neater.

6. PLOS authors have the option to publish the peer review history of their article (what does this mean?). If published, this will include your full peer review and any attached files.

Reviewer #1: No

Reviewer #2: No

---

## [Author Response · Author response to Decision Letter 0]

9 Aug 2023

Response letter

Dear editor:

We are very glad to submit our revision titled “Social capital, depressive symptomatology, and frailty among older adults in the Western areas of China” to your journal. We amended the funding statement as follows: this work was supported by the National Natural Science Foundation of China (grant number 81860599), and addressed all the comments from the reviewers point by point. 

Your sincerely

Zhizhong Wang, Ph.D.

Comments from the Editor

1.Please ensure that your manuscript meets PLOS ONE's style requirements, including those for file naming. 

Response: Yes, we have revised the whole article along with PLOS ONE's style requirements.

https://www.longdom.org/open-access-pdfs/a-confusing-general-term-frailty-should-be-organized-in-relation-with-frieds-criteria-for-frailty-locomotive-syndrome-mu.pdf

https://www.researchgate.net/publication/310021666_Social_capital_interventions_targeting_older_people_and_their_impact_on_health_A_systematic_review

https://bmcgeriatr.biomedcentral.com/articles/10.1186/s12877-022-02815-z

In your revision ensure you cite all your sources (including your own works), and quote or rephrase any duplicated text outside the methods section. Further consideration is dependent on these concerns being addressed.

Response: As the reviewers suggested, we have cited the previous studies, and re-phrased the sentences through whole manuscript. All the changes have highlight in red color.

Response: As we described in the funding part, “this work was supported by the National Natural Science Foundation of China (grant number 81860599)”. The funder had no role in study design, data collection and analysis, decision to publish, or preparation of the manuscript.

4.Thank you for stating the following financial disclosure:

"The authors received no specific funding for this work."

Response: we amended the funding statement within our cover letter as well as in the main text.

5. We are unable to open your Supporting Information file [database.sav]. Please kindly revise as necessary and re-upload.

Response: Now we provided the supporting Information file as EXCEL format.

6. Please include captions for your Supporting Information files at the end of your manuscript, and update any in-text citations to match accordingly. Please see our Supporting Information guidelines for more 

Response: We added a title named Supplementary Table 1. original data at the top of our supplementary file (in the first row), and cited it in body manuscript, described it as “Thus, 2,591 subjects (Supplementary Table 1) were included in the final analysis via a face-to-face structural questionnaire survey ”, can be seen in Participants part, line 100-102.

Reviewer’s Comments to the Author

Reviewer #1

1. Pay attention to the tense and grammar of the article in writing.

Response: We have checked the grammar error over the whole manuscript by a English speaker. And all the changes have highlight in red color. 

2. In Table 2, "After controlling for covariates, SC was negatively correlated with frailty (r=0.07, P=0. 001) and SC was also negatively correlated with depressive symptoms (r=-0.08, P<0.001)", the correlation coefficient seems to be too small for the authors to re-verify the statistical significance.

Response: We have checked the data and re-runed the analysis again, the statistical significance persist, we agree the small effect size, and discussed this in the maintext. 

3. Line 190, r = -0.07 instead of 0.07.

Response: We have revised it and marked in red.

4. The discussion section authors should focus on how depression plays a moderating role in social capital and debilitation.

Response: We supplement the description of moderating role of depression in social capital and debilitation, which can be seen in the last paragraph of discussion part, and marked in red.

Reviewer #2

1. Abstract: Line 44~45: ” SC was identified as a protective predictor of frailty in Chinese old(er) adults in this study…” The authors need to be very careful of making this conclusion because it just shows association but not causality. Does this conclusion overreach?

Response: we revised the description as “SC was identified as a possible protective predictor of frailty in……”.

2. Participants: Line 100: Sample numbers were inconsistent. (2681-90 (42+22+26) = 2591)

Response: We revised it and marked in red, Totally, 2,681 community residents enrolled and finished the face to face survey, of them, 64 were excluded due to the missing value of frailty scale (n=42), and PHQ-9 (n= 22), and 26 aged less then 60 were excluded. Thus, 2,591 subjects (Supplementary Table 1) were included in the final analysis. 

3.Results: Line 199~203: Mediation effect of depression on the relationship of SC and frailty was not explained clearly enough. Is the statistical analysis adequate? Is there any possible confounder?

Response: we revised the description of mediation effect of depression on the relationship of SC and frailty, now described as “after controlling for covariates, there is a significant mediation effect of depression on the relationship between SC and frailty. The total effect SC on frailty was significant (total effect, β = -0.011; 95% CI = -0.017,– 0.005; P = 0.007). The estimated β (95% CIs and P value) of a significant indirect effect mediated by depression was -0.004 (95% CI = -1.006,–0.002 and P = 0.039). The mediation effect explained 36.4% (-0.004/-0.011) of the total variance”. The statistical analysis is adequate, and we controlled the possible confounders includes age, gender, marital status, educational attainment, sleep quality, health behaviors and physical health as described in methods. 

4. “Older adults” is better wording than “old adults”, it should be constantly the same all through the paper.

Response: we changed “old adults” into “older adults” among the whole article.

5. English editing and grammatical adjustment are required.

Response: We have checked the grammar error over the whole manuscript by a English speaker. And all the changes have highlight in red color.

6. Please rearrange Table 1 and Table 3 to be shorter and neater.

Response: we have re-arranged the Table 1 and Table 3, now they are shorter and cleaner.

---

## [Decision Letter · Decision Letter 1]

18 Sep 2023

Social capital, depressive symptomatology, and frailty among older adults in the Western areas of China

PONE-D-23-09139R1

Dear Dr. Wang,

We’re pleased to inform you that your manuscript has been judged scientifically suitable for publication and will be formally accepted for publication once it meets all outstanding technical requirements.

Kind regards,

Wei-Min Chu

Academic Editor

PLOS ONE

Additional Editor Comments (optional):

Reviewers' comments:

Reviewer's Responses to Questions

**Comments to the Author**

1. If the authors have adequately addressed your comments raised in a previous round of review and you feel that this manuscript is now acceptable for publication, you may indicate that here to bypass the “Comments to the Author” section, enter your conflict of interest statement in the “Confidential to Editor” section, and submit your "Accept" recommendation.

Reviewer #1: All comments have been addressed

Reviewer #2: All comments have been addressed

2. Is the manuscript technically sound, and do the data support the conclusions?

Reviewer #1: Yes

Reviewer #2: Yes

3. Has the statistical analysis been performed appropriately and rigorously? 

Reviewer #1: Yes

Reviewer #2: Yes

4. Have the authors made all data underlying the findings in their manuscript fully available?

Reviewer #1: Yes

Reviewer #2: Yes

5. Is the manuscript presented in an intelligible fashion and written in standard English?

Reviewer #1: Yes

Reviewer #2: Yes

6. Review Comments to the Author

Reviewer #1: (No Response)

Reviewer #2: The manuscript has been revised properly according to the reviwers' recommentations; it is now acceptable for publication.

7. PLOS authors have the option to publish the peer review history of their article (what does this mean?). If published, this will include your full peer review and any attached files.

Reviewer #1: No

Reviewer #2: No

---

## [Editor Report · Acceptance letter]

25 Sep 2023

PONE-D-23-09139R1 

Social capital, depressive symptomatology, and frailty among older adults in the Western areas of China 

Dear Dr. Wang:

I'm pleased to inform you that your manuscript has been deemed suitable for publication in PLOS ONE. Congratulations! Your manuscript is now with our production department. 

Kind regards, 

on behalf of

Dr. Wei-Min Chu 

Academic Editor

PLOS ONE